# Diffusive kinks turn kirigami into machines

**Shahram Janbaz** [1] **& Corentin Coulais** [1] ✉

Kinks define boundaries between distinct configurations of a material. In the context of mechanical metamaterials, kinks have recently been shown to underpin logic, shape-changing and locomotion functionalities. So far such kinks propagate by virtue of inertia or of an external load. Here, we discover the emergence of propagating kinks in purely dissipative kirigami. To this end, we create kirigami that shape-change into different textures depending on how fast they are stretched. We find that if we stretch fast and wait, the viscoelastic kirigami can eventually snap from one texture to another. Crucially, such a snapping instability occurs in a sequence and a propagating diffusive kink emerges. As such, it mimics the slow sequential folding observed in biological systems, e.g., Mimosa Pudica. We finally demonstrate that diffusive kinks can be harnessed for basic machine-like functionalities, such as sensing, dynamic shape morphing, transport and manipulation of objects.

Kinks are met across a wide range of scales in materials science, from ferro-electrics and shape-memory alloys undergoing phase transitions[1-3] to flexible metamaterials undergoing mechanical instabilities[3-9]. Such metamaterials are typically made of beams in series[10,11], kirigami[12], hinged mechanisms[13] and inflatable structures[14]. Controlling the motion of kinks in metamaterials offers the fascinating prospect of achieving on-demand mechanical tasks, such as logic[10,15,16], locomotion[6] and shape-changing[12]. Yet, in mechanics the mechanism by which kinks move is either inertia[4-6,10,12,17] or external loading[14,18-21]. In the situation where motion is overdamped, it is hence not possible to observe a traveling kink without constantly loading the material. There are however exceptions if the material is stimuli-responsive, which is commonplace in nature. Plants respond to humidity or to touch[22-25]. A noteworthy example is that of Mimosa Pudica[26-29]: when touching one of its leaves, its leaflets typically sequentially fold up in a few seconds, thus giving rise to a traveling mechanical wave. This is a diffusive kink: the closure and opening of the leaflets can be described as transformation between two stable configurations[30-32] and the propagation of sequential folding is understood to be a reaction-diffusion process[33]. Diffusive kinks are common in the context of chemistry[34,35] and reaction-diffusion is a basic model in theoretical biology such it is used describe the morphogenesis and pattern formations of living systems[36-38], but yet rare in mechanics. Mimosa Pudica is, in fact, the only example that exhibits non-linear diffusive kinks we are aware of. Inspired by this phenomenon, can we engineer a synthetic analog of this mechanical diffusive kink?

A natural avenue to address this question is to use kirigami metamaterials, in which both *geometric* and *dissipative* effects can be leveraged. On one hand, kirigami is a particularly suited platform to exhibit shape-morphing[18,39-41] and in particular kinks[12], controlled by geometry. On the other hand, the use of highly viscoelastic polymers as a constitutive material in combination with buckling has been recently shown to lead to metamaterials with multifunctional dynamic response: their shape-changing response can be tuned on-the-fly by controlling the strain rate[42-45]. Here, we build on these two recent developments to achieve diffusive kinks mimicking that of Mimosa Pudica. We do so in the hitherto unexplored regime of viscoelastic relaxation coupled with buckling of the kirigami. Finally, we use those diffusive kinks to turn kirigami into machines that perform simple mechanical tasks. We believe our findings to be applicable beyond metamaterials, anywhere where first order dynamics dominates and where mechanical instabilities can arise, e.g. poroelasticity[34,46], colloidal assemblies[47], microfluidics controllers[48] and soft robots powered by hydraulics[49,50].

## Multi-texture viscoelastic kirigami

Our approach uses a kirigami, a thick sheet perforated with parallel cutlines, that allows for two local buckling modes—i.e., symmetric and anti-symmetric modes (Fig. 1a), hence two textures[41]. Intrinsically, a single material kirigami is sensitive to geometrical imperfections and the details of the boundary conditions. These factors make the pattern of buckling unpredictable. In our approach, we alleviate this limitation by texturing a thick kirigami with two materials with distinct stiffness

[1]Institute of Physics, Universiteit van Amsterdam, Amsterdam, The Netherlands. ✉e-mail: coulais@uva.nl

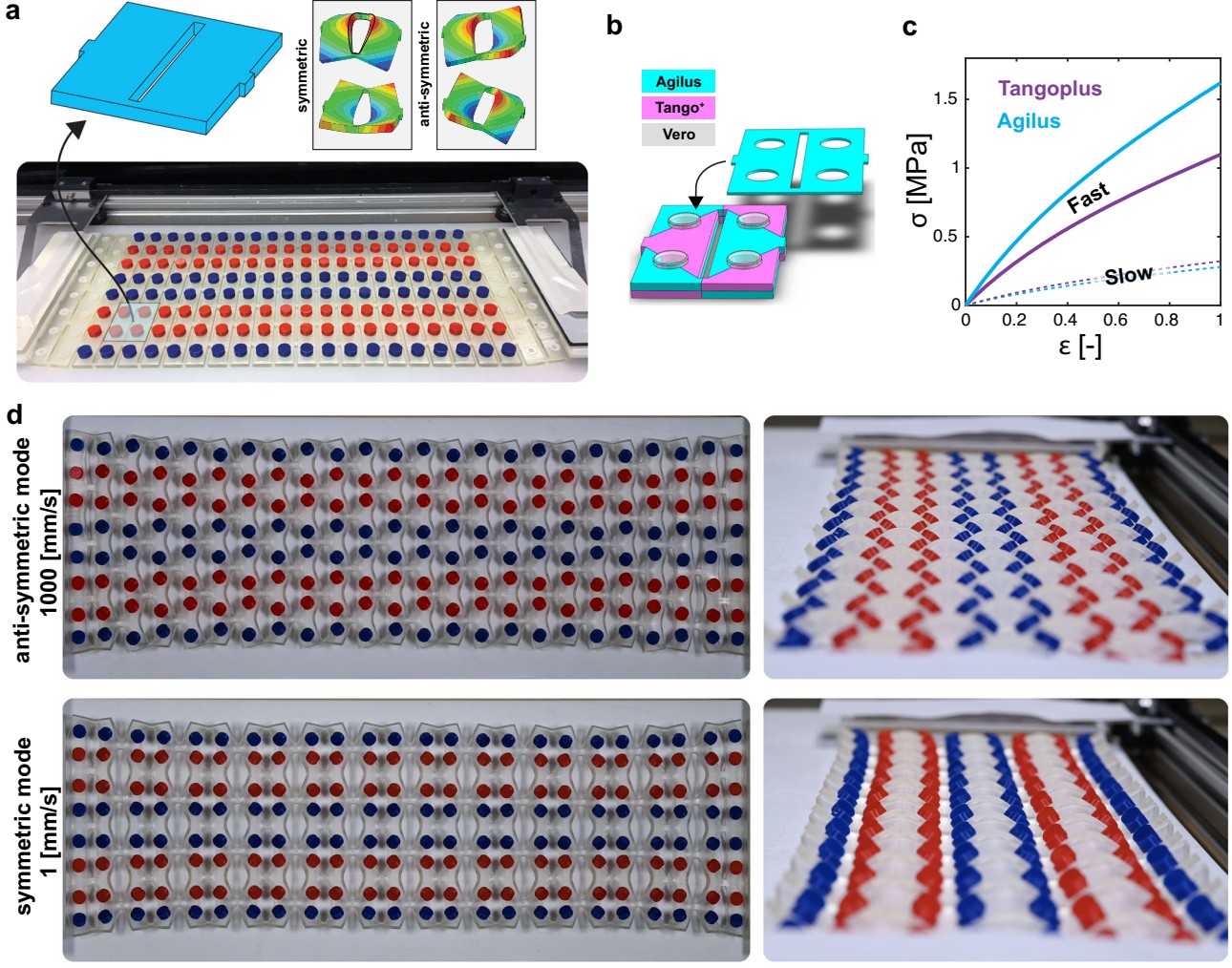

**Fig. 1 | Dynamic control of multi-texture kirigami. a** The behavior of kirigami made by introducing regular parallel cut-lines can be described using one of its representative unit cells. Such unit cells may deform in two (symmetric and anti-symmetric) ways upon stretch. **b** We propose a multi-texture design of kirigami unit cells in order to control the direction of buckling. **c** We use two photopolymers (Agilus and Tangoplus) with a sufficient degree of difference in their viscoelastic properties to print viscoelastic kirigami (see Supplementary Information section B for the advantage of geometrical imperfections). **d** A viscoelastic kirigami made by regular patterning of a TATA unit cell (made of Tango|Agilus|Tango|Agilus, see Supplementary Information section A) exhibits shape-transformations depending on the applied loading rate.

and viscoelasticity. We strategically pattern these two materials in each unit cell through their thickness, using the layout displayed in Fig. 1b. Such patterning enables the control of buckling mode in response to applied loading rates. We use two commercially 3D printable photopolymers (Agilus and TangoPlus, Stratasys) with distinct viscoelastic relaxation strengths to tune the imperfections that control the direction of buckling in response to the applied loading rate[42–45] (Fig. 1c, see Supplementary Information section A, B and C). In fact, one can explain the change in the neutral line along the inner edge of each half unit cell as a result of strain rate. This leads to imperfections that are substantial enough to influence the direction of buckling[51]. We optimized the geometry of the unit cells using a hybrid experimental-computational protocol to ensure the manufacturability of viscoelastic kirigami using polyjet printing[51] and by taking into account the inevitable geometrical imperfections of our 3D printed kirigami, (see Supplementary Information section A). In contrast with single-material kirigami, whose buckling mode is hard to control[41,52], the buckling mode of our multimaterial viscoelastic kirigami is tunable on-the-fly via the rate at which the kirigami is stretched: at low loading rates, the kirigami buckles into a symmetric mode whereas at high loading rates,

it buckles into an anti-symmetric mode, Fig. 1d and Supplementary Video 1).

## Viscoelastic snap-back

Such kirigami exhibits viscoelastic stress relaxation and for some regimes of strain a viscoelastic snap-back (Fig. 2a, b and Supplementary Video 2) is anticipated[45,53,54]. We quantified such snapping by tracking the relaxation of a representative unit cell stretched at a high speed in 3D (see Supplementary Information section D). While at short times, the kirigami unit cell is deformed into the high-speed anti-symmetric mode, at longer times, it gradually deforms into a symmetric mode. Such transformation starts with a slow creep process, but over the course of 100 s, the bottom half of the unit cell snaps back, such that the unit cell ultimately relaxes into the low-speed symmetric mode (Fig. 2c, blue curve). In a limited range, the snap-back of viscoelastic strips is highly sensitive to geometrical imperfections (see Supplementary Information section B). A slight change in the angle of rigid end-connections of the unit cell result in, for example, a shorter delay prior to a viscoelastic snap-back while there is not a visible change in the final angle $\psi$ between the panels (Fig. 2c,

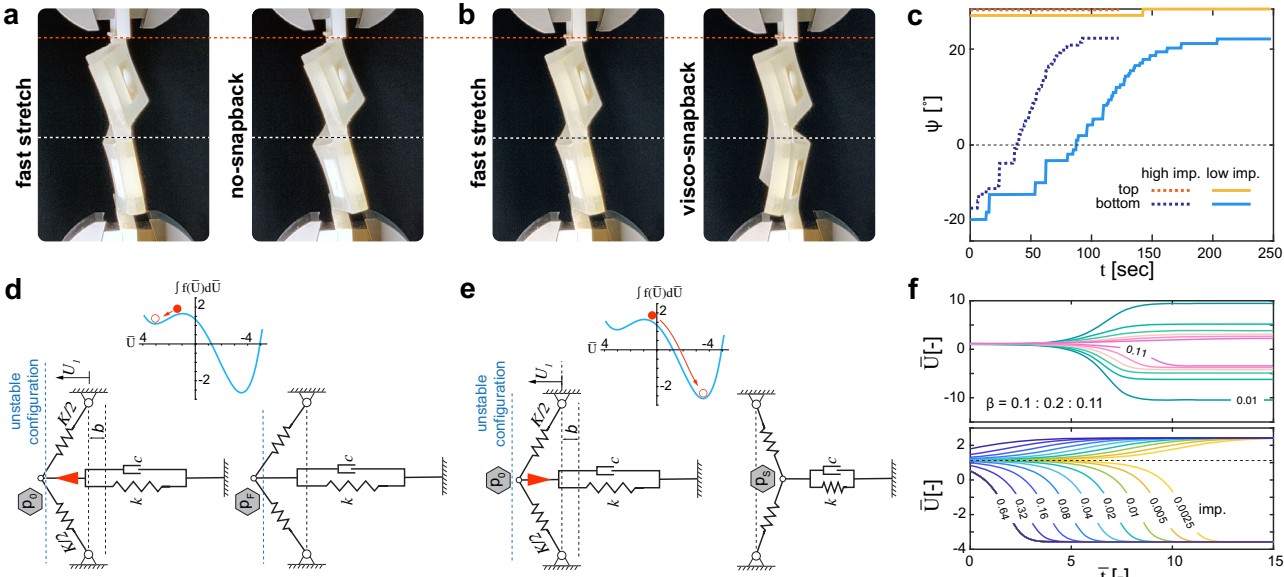

**Fig. 2 | Viscoelastic snapping.** Upon a fast stretch, the geometry of a kirigami unit cell deforms (buckle) anti-symmetrically. **a**, **b** Within certain ranges of stretch, the strip(s) of the buckled unit cell exhibits a viscoelastic snap-back to its lower speed mode−symmetric mode. **c** We used a 3D tracking technique (see Supplementary Information section D) to quantify the role of geometrical imperfections on the acceleration of snap-back. **d**, **e** We use a viscoelastic von Mises truss configuration to represent the condition for the snap-back of a kirigami uni cell from an unstable high-speed to a lower-speed mode (see Supplementary Information section K). **f** Numerically solving the equation of motion (1), regardless of the effect of the initial condition (i.e., geometrical imperfection), shows that the amplitude of snap-back is bonded to the system parameter $\beta$. On the other side, the level of imperfection highly influences the time period prior to a viscoelastic snap-back.

dashed curve). That shows the strong influence of geometrical imperfections[43,53] on the viscoelastic snap-through of our kirigami.

To capture these observations and quantify the effect of material, geometry, and imperfections on viscoelastic snapping, we construct a viscoelastic von Mises truss (Fig. 2d, e) to model switching between symmetric and anti-symmetric buckling modes. The truss is made by jointing a pair of linear springs $K/2$ and a pair of parallel spring $k$ and dashpot $c$ while all the springs are pre-strained. Using a Lagrangian formulation that comprises the elastic potential energy $V$ and the Rayleigh dissipation function $D$[55], we obtain the non-dimensional equation of motion (see Supplementary Information section K):

$$\frac{\partial \bar{U}_1}{\partial \bar{t}} = \bar{U}_1 - \beta \bar{U}_1^3 - 1, \tag{1}$$

in which $\bar{U}_1$ and $\bar{t}$ are the dimensionless displacement and time. The dimensionless parameter $\beta = \frac{Kk^2b^2d_0}{2((-K-k)a+Kd_0)^3}$ is a function of material properties and nonlinear geometry, where $d_0$ is the natural length of the springs $K/2$, $a$ is half of the distance between the fixed joints, and $b$ is the offset point $P$ from and straight configuration while the spring $k$ is relaxed. Note that related models have been considered in the literature in slightly different contexts[53,54,56], see ref. 45 for a recent review. Although such model is a much simplified version of the kirigami that exhibits complex 3D deformations, one can still build an intuitive connection between the two. The pair of springs combined with the viscoelastic element, i.e., dashpot and springs, can describe the transition from an anti-symmetric to a symmetric mode. Also, one can intuitively relate the parameter $\beta$ to the level of prestretch of the kirigami: the more the pre-stretch, the more the deflection of the kirigami and hence the larger $\beta$. This is of course merely a qualitative mapping, yet which describes the main features of relaxation dynamics of the kirigami.

The right-hand side of Eq. (1), $f(\bar{U}_1) = \bar{U}_1 - \beta \bar{U}_1^3 - 1$, determines whether the system is monostable ($\beta < 0$ or $\beta > \frac{4}{27}$) or metastable ($0 < \beta < \frac{4}{27}$)[57]. In the latter case, the system will snap or not depending on the initial condition, viz. the offset from the unstable configuration

(Fig. 2d, e). The parameter $\beta$ then not only determines whether the unit cells snaps, it also determines the magnitude of snap-back (Fig. 2f): the smaller $\beta$, the larger the magnitude of snapping. In contrast, the level of imperfection determines the time it takes for snapping to occur (Fig. 2f): the more the imperfection, the shorter the initial creep and the quicker the snapping. Qualitatively, the viscoelastic snapping of a unit cell can be described as follows: the unit cell is metastable. It can remain trapped in the anti-symmetric state or snap into the symmetric state. What determines whether it snaps or not is the direction of imperfection perturbing from the unstable configuration. For larger pre-stretch, this imperfection is mostly guided to the high-speed mode and the unit cell does not snap (Fig. 2a, c), while for lower pre-stretch, this imperfection is relatively larger and towards the low-speed mode so the unit cell does snap (Fig. 2b, e).

## Emergence of diffusive kinks

We then hypothesize that similar to the propagation of inertial kinks in bi-stable chains[5], our viscoelastic trusses in series will exhibit kinks, but we expect the dynamics of these kinks to be diffusive[58]. To examine this assumption, we assemble unit cells of Eq. (1) to a one-dimensional strip. We, therefore, model a viscoelastic kirigami strip as lumped viscoelastic trusses, connected by linear springs $R$ (Fig. 3a). The dimensionless equation of motion, excluding the effect of inertial forces, in continuum limit can be then expressed as (see Supplementary Information section K):

$$\frac{\partial \bar{U}}{\partial \bar{t}} = \bar{U} - \beta \bar{U}^3 - 1 + \frac{\partial^2 \bar{U}}{\partial \bar{X}^2} \tag{2}$$

in which $\bar{X}$ is the nondimensionalized coordinate. Eq. (2) is a reaction-diffusion equation that is normally used to describe transitions in metastable media in chemistry[59] and are well known to host diffusive kinks, but that describes in our case a mechanical analog.

Numerical solutions of Eq. (2) confirm the emergence of a kink following the transition between the initial metastable up-state ($\bar{U}_+$) and the stable down-state ($\bar{U}_-$) (Fig. 3b, c). Alternating the initial

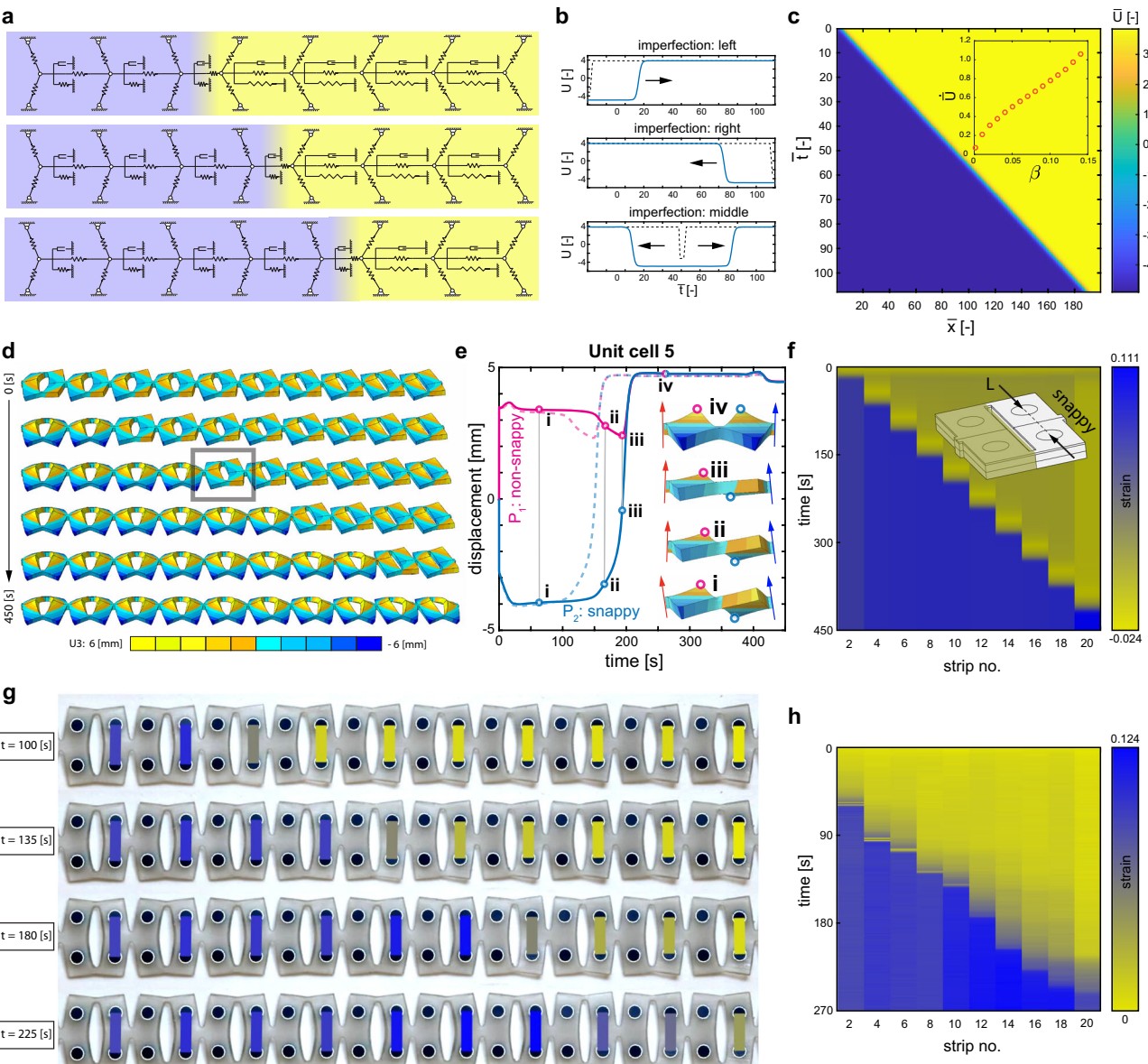

**Fig. 3 | Diffusive kinks in pre-stretched kirigami strips. a** We use a series of viscoelastic trusses integrated using linear springs $R$ to represent a kirigami strip. **b** Numerically solving the reaction-diffusion equation (2) for $\beta = 0.05$ reveals the emergence of a kink due to the transition from high- and low-speed configurations of the individual viscoelastic trusses. The direction of traveling kinks depends on the initial imperfections. The high- and low-speed configurations of the individual viscoelastic trusses correspond to the maximum and minimum roots of the reaction function $f(\bar{U}) = \bar{U}_1 - \beta\bar{U}_1^3 - 1$ respectively (i.e., $\bar{U} \approx +3.843$ and $-4.907$). **c** Independent from the value of $\beta$, the resultant kink travels at a constant speed. The inset shows the velocity of the kink increases as the parameter $\beta$ reaches its highest bound which corresponds to the lowest difference between high- and low-speed configurations. the parameter $\beta$. **d** Our computational models of kirigami

strips confirm that such kinks can be observed in viscoelastic kirigami. See Methods and Supplementary Information section B for details. **e** Following the behavior of unit cell *five*, tracking the movement of the points $P_1$ and $P_2$ on the symmetry line of the non-snappy and snappy half unit cells over time, reveals that the prerequisite condition for the snap-back is the geometrical imperfection that is transferred via unit cell *four*. **f** Measuring the lateral staining of the snappy half unit cells, even half unit cells, starting from a high-speed anti-symmetric mode, shows that the resultant kink travels at a constant speed. **g** A kirigami strip consisting of twelve identical unit cells exhibits a traveling kink (see Supplementary Information section E). **h** Measuring the values of lateral strains, corresponding to the snapping of the even index strips, confirms the emergence of a constant speed kink.

conditions demonstrate that the emergence and the propagation of diffusive kinks can be understood as the propagation of an imperfection (Fig. 3b and Supplementary Video 3). With small imperfections, no kink is nucleated. If the imperfection is large enough to make a unit cell snap, a kink is nucleated and propagates at a constant velocity (Fig. 3c). The kink's velocity increases with the parameter $\beta$ (inset Fig. 3c). The larger the energy difference between the energy of the up-state and of the down-state, the faster the kink.

In order to take all nonlinearities of our real kirigami into account, we employed nonlinear computational mechanics using a commercial finite element method software (Abaqus, 2020). Our model allows us to capture the primary step of fast stretching leading to the high-speed mode associated with two commercially available photopolymers (i.e., Agilus and TangoPlus) that we use in our experiments. However, Agilus and TangoPlus exhibit poor contrast in their viscoelastic properties (see Supplementary Information section B) and are not solely sufficient to stimulate a snap-back and, therefore, traveling kink. We, thus,

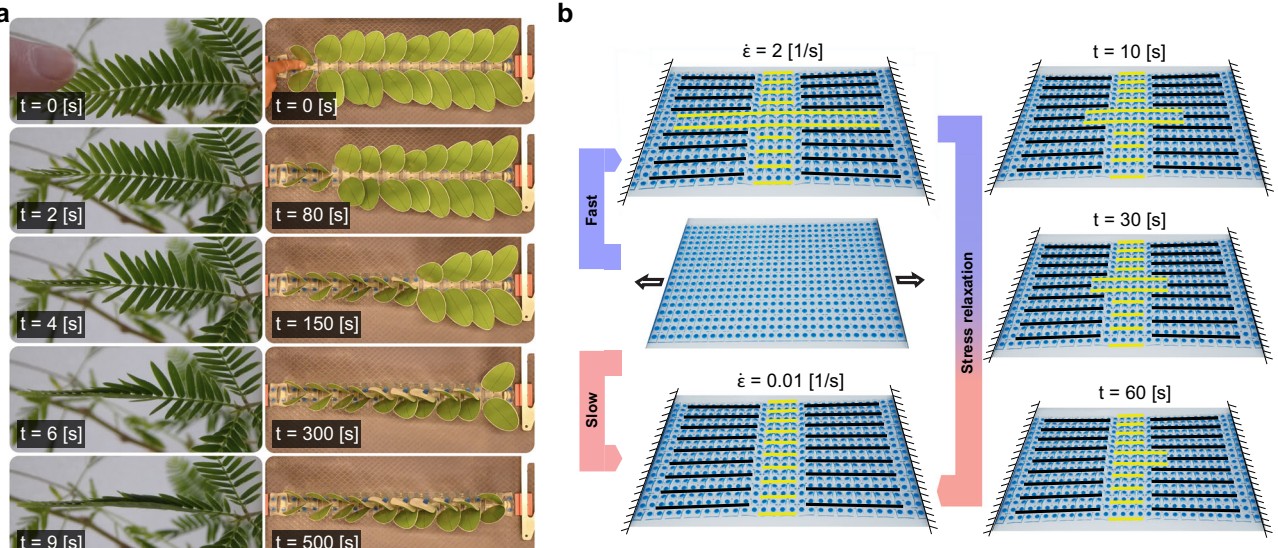

**Fig. 4 | Sensing and shape-morphability. a** The leaflets of a Mimosa Pudica plant sense their environment by exhibiting a sequential folding process once touched. **b** A strain-rate dependent 2D shape-morphable plate exhibits dynamic shape transformation from its high-speed to low-speed mode.

benefit from the geometrical imperfections that arise inevitably from polyjet printing of soft kirigami to overcome this limit[51] (see methods). Consequently, we begin with a strip of unit cells with broken geometrical symmetry. As anticipated, the snappy half unit cells exhibit snap-back once they are pushed from their high-speed meta-stable configuration. Over-stretching then prevent the simultaneous snap-back of all unit cells and the boundary of the non-snappy half unit cell triggers the traveling of a dissipative kink (Fig. 3d and Supplementary Video 4) while it is prevented at the other side. We can justify the traveling of such a kink as the result of imperfection transfer through the boundaries of the consecutive unit cells.

To quantify this, let us look at the snap back of the unit cell *five* (indexed from the left side). Tracking the movement of the points $P_1$ and $P_2$ on the symmetry line of the non-snappy and snappy half unit cells (Fig. 3e) we observe a trend of deformation towards a flat configuration prior to a snap back. In fact, this trend is a prerequisite for the snap-back in the unit cells of kirigami which is not permitted if the previous unit cell is still in its anti-symmetric (high-speed) mode (Fig. 3e(i)). The snap-back of the unit cell *five* is kicked off by a negative (clockwise) rotation at its boundary with unit cell *four*−coincident with the final stage of mode transformation of the unit cell *four* (Fig. 3e(ii,iii)). As soon as the transition from anti-symmetric to symmetric mode is complete, the left boundary of the unit cell *five* rotates back to its positive angle while the right boundary rotates negatively (blue arrow in Fig. 3e(iv)). Therefore, the propagation of the kink is triggered by the transmission of a geometric imperfection from one unit cell to the next. We further measured the lateral strain of the snappy half unit cells over time to evaluate the propagation of diffusive kinks. Our nonlinear model then confirms the constant speed propagation of diffusive kink in viscoelastic kirigami (Fig. 3f) while a constant force boundary condition is set (see Supplementary Information section B).

To examine whether such a wave is achievable experimentally, we build a kirigami strip, prestretch it fast such that it buckles into the anti-symmetric mode and let it relax (see Supplementary Information section A). Note that significant geometric imperfections, akin to those introduced in our computational model, have been observed in the multijet 3D printed specimens. Our experiments confirm the emergence of a localized mechanical kink starting from the high-speed anti-symmetric mode, into the low-speed symmetric mode (Fig. 3g and Supplementary Video 5. The mechanical kink travels all the way from one side to the other side of the kirigami. The nucleation of the kink is

related to the imperfection that arises from the clamping condition at one end-side, while at the other end it is prevented.

To quantify the dynamics of such a slow kink, we measured the lateral strain of the half unit cell (See Supplementary Information section E, Supplementary Video 6), which undergoes a jump upon snapping (Fig. 3h). The sequence of such snapping events reveals the existence of a kink, traveling at constant velocity (around 1 *mm/s*). Clearly, the timescale is much larger than the speed of vibrations and the slow kink undergoes purely diffusive dynamics. We note that the nonhomogeneous frictional forces between the substrate and the kirigami structures and the geometrical imperfections from fabrication influence the traveling of kinks.

Our models, in combination with the experiments, provides the following physical picture: the time delay before the first unit cell snaps (i.e., corresponding to strip no. 2) can be attributed to the size of the imperfection at the boundary while the other unit cells are still slowly creeping in their high-speed anti-symmetric mode (Fig. 3h). When the first unit cell snaps, it creates a larger imperfection for its neighbor and hence accelerates its snapping. This snapping of this second unit cell will in turn create a larger imperfection and accelerate the snapping of the third unit cell, and so on. This sequential mechanism ultimately creates a traveling kink with purely overdamped dynamics: a diffusive kink.

## Shape-transformation via a touch

If there is no imperfection, no kink will therefore be nucleated. Inspired by the sequential folding of Mimosa Pudica (Fig. 4a, Supplementary Video 7), we can however externally load the metamaterial to induce such imperfection. Just as in the case of Mimosa Pudica, if the load is large enough, a diffusive kink will emerge and traverse the sample (Fig. 4b, Supplementary Video 7). Such diffusive kink could hence potentially be used as a functional principle for sensing applications[60].

We now show that such principle can also be applied to obtain dynamic shape-changing functionalities in 2D[61,62]. To do this, we modulate the strain-rate dependency and the snapping properties of kirigami unit cells by slightly varying their geometry and by rationally patterning the viscoelastic material in the kirigami (see Supplementary Information section F), such that the high-speed mode of the kirigami is in the shape of a "plus" sign and the low-speed more in the shape of a "minus" sign. We then confirm experimentally that this kirigami exhibits a dynamic shape-transformation, from a plus- to a minus-textured

pattern, subsequent to a fast stretch (Fig. 4b, Supplementary Video 8). In our experiments, we observe a progressive shape transformation, starting from the two clamped sites of the kirigami plate, that results in merging of the textures at the two side of the minus strip. This transformation is mediated by a slow dissipative kink described above. Finally, the kirigami adopts a minus sign texture which is the same as what we observed in the slow-speed experiments. A kirigami with such time-dependent shape transformation can potentially be miniaturized to a platform for designing metamaterials with dynamic physical properties such as dynamic holograms[63].

### Equipped with kinematic chains viscoelastic kirigami manipulated objects in non-time reversal cycles

We demonstrate that this diffusive kink can also be used to perform basic mechanical tasks. First, we show that it can be used to transport an object forward[64]. For this purpose, we dress a kirigami strip with arm-links and guiders (Fig. 5a and S8, Supplementary Video 9, and see Supplementary Information section H). As described above, such kirigami, after being stretched at a high speed and set to relax, exhibits a diffusive kink. The diffusive kink lifts the arm-links and guiders sequentially and carries an object—in this case a ping-pong ball—forward.

To further illustrate the capability of viscoelastic kirigami to manipulate objects, we produced a kirigami with three unit cells that each carry stiff and flexible arms (see Supplementary Information section I). Upon a fast stretch, the kirigami buckles into its asymmetric mode. Hence, the arms are deflected towards each other and each unit cell is able to hold a ping-pong ball (Fig. 5b and S9, Supplementary Video 10). Over time, a diffusive kink emerges and the kirigami sequentially snaps, resulting in a sequential release of the ping-pong balls.

Finally, we demonstrate the ability of the diffusive kink to do work on its environment. We do so by integrating on each unit cell two compliant four-bar linkages, each connected to a paddle (Fig. 5c). We see that upon a cycle of stretching and snapping instability of the kirigami, each paddle moves in a cycle (Fig. 5d). Cyclic motion is a critical requirement of many soft robotic devices that need to break time-reversal symmetry to do work. When put together in a larger kirigami, we can simulate a device that, once it is being stretched, rows in sequence as the diffusive kink propagates (Fig. 5e, Supplementary Video 11). Since each paddle performs a cyclic motion, such kirigami could be used for locomotion or excavation. Hence the combination of diffusive kinks with cycles could further enrich the dynamical functionalities of soft robots[14,65].

The forces deployed by our kirigami are intrinsically bounded by the contrast in the viscoelastic properties of the materials we use. We envision that advances in polymer science and 3D printing technology will help push those bounds further and make diffusive kinks more robust. We envision the fast-paced development in additive manufacturing will soon give rise to polymers with extreme contrast in their instantaneous and long-term stiffness values. Such materials with giant viscoelascity tunability will dramatically empower machines based on diffusive kinks, viz. allow them to carry heavy loads.

### Summary

In summary, we have shown that viscoelasticity in kirigami structures can be harnessed to create bistable mechanisms, whose snapping is mediated by creep and that in turn exhibit diffusive traveling kinks. These diffusive kinks can be used to power kirigami machines that exhibit basic mechanical tasks, such as sensing, dynamic 2D shape morphing and object manipulation. In contrast to inertial kinks[3-6,12], diffusive kinks can propagate arbitrarily slowly. These features facilitate the design of materials that mimic the motion of plants[59], and hence add a concept to the toolbox of soft robots[66,67].

## Methods
### Design

In our study, we employed a simple pattern of parallel cut lines to shape kirigami with periodic unit cells. To attain the capability for

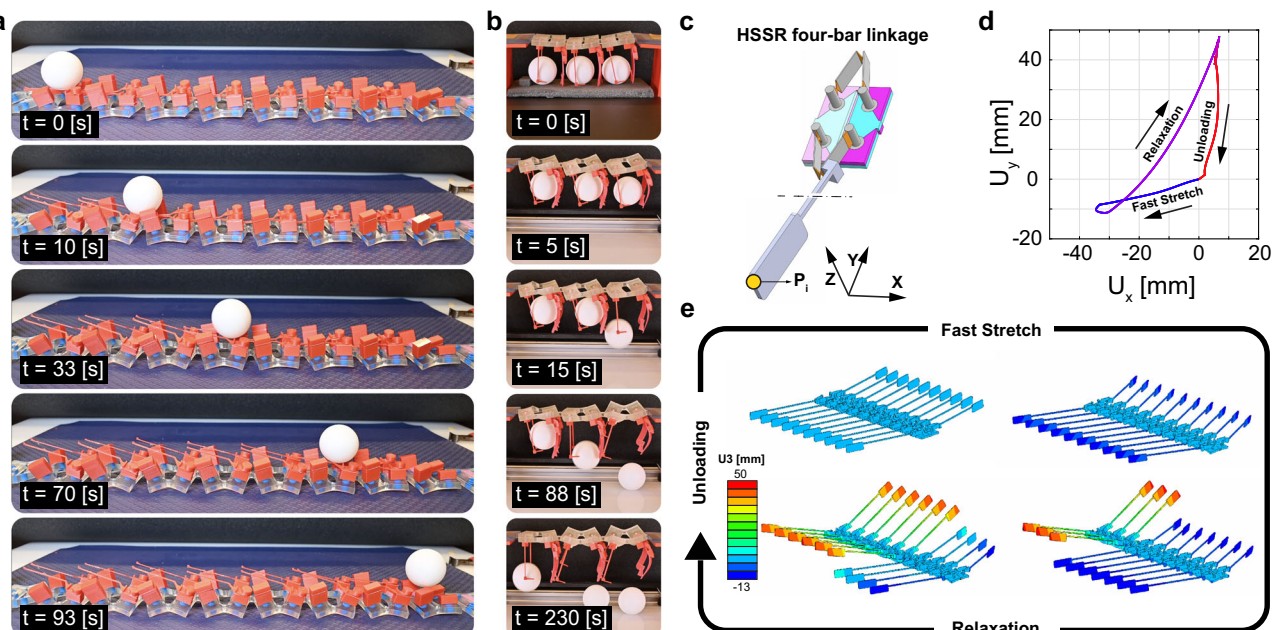

**Fig. 5 | Turning kirigami into machines. a** A kirigami strip equipped with arm-links and guiders carries a ping-pong ball while initially stretched at a high speed. **b** Alternatively functionalized, a kirigami strip consists of three unit cells equipped with rigid arms and two-pole kirigami grippers demonstrates instantaneous grasp and sequential release of ping-pong balls. **c** Kirigami unit cells can carry spatial mechanisms to extend their functionalities. **d** Following the point $P_i$ in *XZ* plane show how that viscoelastic snap back in combination with the four bar linkage induces a cyclic motion. **e** The computational modeling of a kirigami strip carrying paddles using integrated spatial four-bar linkages shows an example of soft robotic devices that sequentially moves its component on cyclic paths as a consequence of the diffusive kink.

strain rate multi-functionality, we selected thick unit cells, partitioned them to accommodate multi-materials, and adjusted their dimensions to ensure alignment with computational predictions experimentally. While the viscoelastic materials used for printing flat kirigami can steer high-speed and low-speed buckling, their properties are insufficient for conducting snap-back in pre-stretched unit cells. The geometry of kirigami can then be adjusted by introducing purposeful geometrical imperfections to demonstrate viscoelastic snap-back—interestingly such imperfections are the inevitable consequence of polyjet 3D printing of soft photopolymer. To evaluate this hypothesis, we introduced such geometrical imperfections in our computational model. We observed that if those imperfections are large enough, snap-back and, eventually, non-linear waves occur (see Supplementary Information section A). These observations are consistent with the experimental findings and confirm the important role of imperfections for diffusive kinks.

### Manufacturing and experimental study

Polyjet printing is a technology that is capable of fabricating complex multimaterial geometries by jetting and curing various photopolymer[68]. We used two commercially available soft photopolymers (i.e., Agilus and TangoPlus) and a rigid photopolymer to fabricate viscoelastic kirigami using a commercial 3D printer (Objet500 Connex3, Stratasys, see Supplementary Information section A, B and C). To reach an effective strain rate and achieve high-speed mode of buckling, we developed a costume-made test bench with a low-friction Teflon substrate allowing us to adjust the speed of stretch in the range of 1 to 5000 mm/s. We then analyzed our experiments by tracking the embedded particles (i.e., black dots and monochrome patterns) on the surface of kirigams using Matlab codes and analyzed the snap-back of kirigami unit cells in 3D using a Python code (see Supplementary Information section D).

### Computational modeling

In order to analyze the behavior of multi-functional viscoelastic kirigami, we used non-linear computational mechanics (Abaqus 2020, standard solver). We used a visco-hyperelastic material model (using the first term of Prony series) to define the visco-hyperelasticity of Agilus and TangoPlus and an elastic material model to define rigid parts. The viscoelastic material parameters were determined by minimizing the difference between the stress values predicted by the material model and the experimental data, assuming that the two viscoelastic polymers are incompressible[69] (see Supplementary Information section B). To discretize the geometry of the kirigami, we used three-dimensional elements C3D8H. A mesh convergence study has been performed to ensure the insensitivity of our computational analysis to the mesh size. The clamped-clamped condition has been realized by fixing the nodal points of one end of the kirigami while the nodal points of the moving end have the freedom to move along the axial direction of the plates. In some models, we tied the end nodal points at each end to a reference point. Periodic boundary condition has been satisfied by constraining the motion of the nodal points at the two free boundaries of the unit cells with respect to a reference point placed on one node. To model the propagation of diffusive kink in kirigami, we introduced geometrical imperfections, similar to those observed in experiments, based on a preliminary static simulation (see Supplementary Information section C).

### Analytical modeling

We built a reduced model based on a viscoelastic von Mises truss to model the shape-transformation of individual viscoelastic kirigami unit cells. We then assembled identical trusses using identical linear springs to drive the equation of motion of such a system and explore the wave propagation in viscoelastic kirigami in the continuum limit. Our model is detailed in Supplementary Information, section K.

## Data availability

All the data supporting this study are available on the public repository https://zenodo.org/doi/10.5281/zenodo.10632577.

## Code availability

All the codes supporting this study are available on the public repository https://zenodo.org/doi/10.5281/zenodo.10632577.

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

## Acknowledgements

We thank Freek van Gorp, Jonas Veenstra, Keivan Narooei and Mazi Jalaal for insightful discussions, Anastasiia Krushynska for real-time imaging of Mimosa Pudica, Clint Ederveen Janssen, Daan Giesen, Kasper van Nieuwland, and Ronald Kortekaas for their technical support. We acknowledge funding from the Netherlands Organisation for Scientific Research under grant agreement NWO TTW 17883 and from the European Research Council under grant agreement 852587.

## Author contributions

S.J. and C.C. designed the study. S.J. executed the experiments and performed computational and analytical analysis, and C.C. supervised the project. S.J. and C.C. wrote the manuscript.

## Competing interests

The authors declare no competing interests.
