## [Peer Review File · Nature Communications]

Diffusive kinks turn kirigami into machinesEditorial Note: This manuscript has been previously reviewed at another journal that is not operating a transparent peer review scheme. This document only contains reviewer comments and rebuttal letters for versions considered at Nature Communications.

REVIEWER COMMENTS

Reviewer #1 (Remarks to the Author):

This manuscript delves into the phenomena of propagating kinks in purely dissipative kirigami structures and their potential utility in simulating basic mechanical functions. The authors engineered kirigami configurations that undergo texture transitions based on the velocity of their stretching. Swift stretching of the viscoelastic kirigami followed by a period of rest can trigger a sudden transition between textures, giving rise to a propagating diffusive kink. This research underscores the potential of diffusive kinks in applications such as sensing, dynamic morphological changes, and the transportation and manipulation of objects. After reading through the entire article, I think it has some originality. However, major revisions are still necessary. My primary concerns are listed as follows.

1. The authors posited that out-of-plane buckling instabilities in kirigami structures present in both symmetric and anti-symmetric forms, and that the buckling modes are random. This is evidently incorrect. There are numerous reports suggesting that the buckling modes are sensitive to defects, geometry, and boundaries. Thus, I have reason to believe that the authors did not conduct a comprehensive parametric study.
2. Regarding the multi-material 3D printing mentioned in the manuscript, the fabrication method and its relevance to the main theme have not been clearly illustrated. As a reader, I find it rather perplexing. I suggest the authors clarify the relevant applications and features in the revised manuscript. Regarding Figure 1c, the authors adopted a different illustration method, making it difficult for me to readily grasp its deeper implications.
3. To enhance the dissemination of this paper and enable the authors to easily validate the results, I hope the authors can provide the code for solving the differential equations presented in this work (such as MATLAB/Mathematica/Python code). This is because the mechanical model used by the authors does not seem particularly novel.

Reviewer #2 (Remarks to the Author):

The authors worked very hard to address my previous comments and concerns. I am satisfied with the revised work, and think it will make a nice addition to this journal.

REVIEWER COMMENTS

Reviewer #1 (Remarks to the Author):

This manuscript delves into the phenomena of propagating kinks in purely dissipative kirigami structures and their potential utility in simulating basic mechanical functions. The authors engineered kirigami configurations that undergo texture transitions based on the velocity of their stretching. Swift stretching of the viscoelastic kirigami followed by a period of rest can trigger a sudden transition between textures, giving rise to a propagating diffusive kink. This research underscores the potential of diffusive kinks in applications such as sensing, dynamic morphological changes, and the transportation and manipulation of objects. After reading through the entire article, I think it has some originality. However, major revisions are still necessary. My primary concerns are listed as follows.

REPLY: We appreciate your valuable review and the insightful comments you have provided. In the revised version of our manuscript, we have included statements to provide additional clarity regarding the scope of our article and the rationale behind our research approach.

1. The authors posited that out-of-plane buckling instabilities in kirigami structures present in both symmetric and anti-symmetric forms, and that the buckling modes are random. This is evidently incorrect. There are numerous reports suggesting that the buckling modes are sensitive to defects, geometry, and boundaries. Thus, I have reason to believe that the authors did not conduct a comprehensive parametric study.

REPLY: We believe there is a possible misunderstanding that probably stems from terminology. We agree with you: the buckling mode of kirigami is controlled by imperfections. However, those imperfections are often ill-controlled. Hence the buckling mode typically changes between specimens and appears to be random. This poor control is precisely what we alleviate here by introducing patterned viscoelasticity. One can in fact interpret the patterned viscoelasticity as a means to control the imperfection dynamically via the strain rate. In the design process, we have in fact carried out a study of the role of imperfections. We agree however that it was not explained in sufficient detail in the previous manuscript.

REVISION: To provide additional clarification, we have included the following statements in the main text:

“Intrinsically, a single material kirigami is sensitive to geometrical imperfections and the details of the boundary conditions. These factors make the pattern of buckling unpredictable. In our approach, we alleviate this limitation by texturing a thick kirigami with two materials with distinct stiffness and viscoelasticity. We strategically pattern these two materials in each unit cell through their thickness, using the layout displayed in Fig. 1b.”

In addition, we provide a more comprehensive explanation on the way we modelled imperfections in the Methods.

“While the viscoelastic materials used for printing flat kirigami can steer high-speed and low-speed buckling, their properties are insufficient for conducting snap-back in pre-stretched unit cells. The geometry of kirigami can then be adjusted by introducing purposeful geometrical imperfections to demonstrate viscoelastic snap-back---interestingly such imperfections are the inevitable consequence of polyjet 3D printing of soft photopolymer. To evaluate this

hypothesis, we introduced such geometrical imperfections in our computational model. We observed that if those imperfections are large enough, snap-back and, eventually, non-linear waves occur (see ESI). These observations are consistent with the experimental findings and confirm the important role of imperfections for diffusive kinks.”

2. Regarding the multi-material 3D printing mentioned in the manuscript, the fabrication method and its relevance to the main theme have not been clearly illustrated. As a reader, I find it rather perplexing. I suggest the authors clarify the relevant applications and features in the revised manuscript. Regarding Figure 1c, the authors adopted a different illustration method, making it difficult for me to readily grasp its deeper implications.

REPLY: Thank you for your comment. We would like to stress that detailed information on the design of the unit cell and on the manufacturing is explained in detail in the Supplementary Information.

REVISION: It is important to note that our use of two transparent photopolymers posed an illustrative limitation. We mitigate this challenge by providing supplementary videos that clarify our concept. To provide further clarity regarding our 3D printing technique and its implications for applications, we have incorporated the following statement in the Methods:

“Polyjet printing is a technology that is capable of fabricating complex multimaterial geometries by jetting and curing various photopolymer [68]. We used two commercially available soft photopolymers (i.e., Agilus and TangoPlus) and a rigid photopolymer to fabricate viscoelastic kirigami using a commercial 3D printer (Objet500 Connex3, Stratasys, see ESI).”

In addition, as we believe the reviewer one refers to figure 1d (not Fig 1c), we revised it to make it easier to understand.

3. To enhance the dissemination of this paper and enable the authors to easily validate the results, I hope the authors can provide the code for solving the differential equations presented in this work (such as MATLAB/Mathematica/Python code). This is because the mechanical model used by the authors does not seem particularly novel.

REPLY: We have already furnished the MATLAB code that validates our results through the ZENODO link <https://doi.org/10.5281/zenodo.7404494>.

Reviewer #2 (Remarks to the Author):

The authors worked very hard to address my previous comments and concerns. I am satisfied with the revised work, and think it will make a nice addition to this journal.

REPLY: We appreciate your positive feedback, and we are thankful for your review.

REVIEWERS' COMMENTS

Reviewer #1 (Remarks to the Author):

The authors have thoroughly considered each of my comments and concerns, and provided me with comprehensive explanations. I am pleased with the improvements made to the paper, and I think that the current version is suitable for the journal.